# Weight Reduction with GLP-1 Agonists and Paths for Discontinuation While Maintaining Weight Loss

**DOI:** 10.3390/biom15030408

**Published:** 2025-03-13

**Authors:** Allison B. Reiss, Shelly Gulkarov, Raymond Lau, Stanislaw P. Klek, Ankita Srivastava, Heather A. Renna, Joshua De Leon

**Affiliations:** 1Department of Medicine, NYU Grossman Long Island School of Medicine, Mineola, NY 11501, USA; raymond.lau@nyulangone.org (R.L.); stanislaw.klek@nyulangone.org (S.P.K.); joshua.deleon@nyulangone.org (J.D.L.); 2Department of Foundations of Medicine, NYU Grossman Long Island School of Medicine, Mineola, NY 11501, USA; shellygulk1234@gmail.com (S.G.); ankita.srivastava@nyulangone.org (A.S.); heather.renna@nyulangone.org (H.A.R.)

**Keywords:** treating obesity, GLP-1 receptor agonists, body weight, drug therapy, appetite regulation, energy balance, adverse effects

## Abstract

Worldwide, nearly 40% of adults are overweight and 13% are obese. Health consequences of excess weight include cardiovascular diseases, type 2 diabetes, dyslipidemia, and increased mortality. Treating obesity is challenging and calorie restriction often leads to rebound weight gain. Treatments such as bariatric surgery create hesitancy among patients due to their invasiveness. GLP-1 medications have revolutionized weight loss and can reduce body weight in obese patients by between 15% and 25% on average after about 1 year. Their mode of action is to mimic the endogenous GLP-1, an intestinal hormone that regulates glucose metabolism and satiety. However, GLP-1 drugs carry known risks and, since their use for weight loss is recent, may carry unforeseen risks as well. They carry a boxed warning for people with a personal or family history of medullary thyroid carcinoma or multiple endocrine neoplasia syndrome type 2. Gastrointestinal adverse events (nausea, vomiting, diarrhea) are fairly common while pancreatitis and intestinal obstruction are rarer. There may be a loss of lean body mass as well as premature facial aging. A significant disadvantage of using these medications is the high rate of weight regain when they are discontinued. Achieving success with pharmacologic treatment and then weaning to avoid future negative effects would be ideal.

## 1. Introduction

Obesity as defined by The Obesity Society (TOS) is a multi-factorial chronic disease that results from excess fat accumulation that presents a risk to health [1]. Multiple organ systems are affected which leads to additional chronic diseases associated with obesity. Obesity may present itself with multiple clinical phenotypes and also varied treatment responses. These varied treatment responses likely originate from our limited understanding of the mechanisms of weight regulation. The concept that body fat storage may be regulated was first proposed by Kennedy et al. through the concept of a “set point” [2]. He suggested that adipose tissue may produce a signal that may be sensed by the brain to target a “level of body fatness”. This model of body fat regulation was widely adopted in the 1990s with the discovery of leptin [3,4].

The set point theory remains a hypothesis as the molecular mechanisms behind it continue to be ambiguous. This is also why developing pharmacologic treatments is challenging without clear targets. However, in the last 20 years, the gastrointestinal system has been identified as the largest endocrine organ and has demonstrated the key role of gut hormones in energy homeostasis [5]. The discovery of the gut hormone glucagon-like peptide-1 (GLP-1) and the synthesis of agonists for its corresponding receptor (GLP-1 receptor) has tremendously impacted treatment for weight reduction. However, significant questions remain about these drugs. Based upon the “set point” theory of weight regulation, the possibility of needing these medications over extended periods of time to avoid the inevitable weight regain may cause the clinician to consider other treatments. The lack of long-term clinical trials in weight management compels clinicians to consider potentially unforeseen long-term side effects of the GLP-1 receptor agonists. The known risks of pancreatitis, gastroparesis, and lean body mass loss are variables to be considered as well. In order to present these drugs with a balance of their pros and cons, the longer-term studies showing cardiovascular benefits are also taken into account.

In this review, we examine the health risks of obesity and the overall magnitude of the problem. A discussion of potential therapeutic strategies including dietary, physical activity, bariatric surgery, and pharmacologic therapy are described. GLP-1 receptor agonist drug therapy is a key focus with consideration of the mechanism of action, clinical trials in weight management, and the potential role of the drug category in weight maintenance. We aim to provide a balanced discussion of the benefits of GLP-1 receptor agonists, as well as the risks and unknown effects. In an ideal setting, weight loss achieved with GLP-1 agonist therapy in the short-term could be durable without pharmacotherapy for many years and perhaps over a lifetime.

## 2. Methodology

The search methodology employed in this narrative review was comprehensive and aimed to capture current relevant evidence pertaining to GLP-1 medication use for weight loss and discontinuation of these medications. For the purposes of gathering information and preparing this paper, PubMed, Scopus, Embase, Web of Science, and Google Scholar were searched for peer-reviewed literature in English on 8 and 9 October 2024 and again for updates on 25 January 2025. We also scrutinized details of clinical trial protocols within ClinicalTrials.gov. Key terms included “obesity”, “GLP-1 receptor agonists”, “weight regain”, “weight loss”, “liraglutide”, “semaglutide”, “tirzepatide”, “exenatide”, “dulaglutide, “ phentermine/topiramat ”, “contrave”, “bariatric surgery”, “treatment outcomes”, “adverse effects”, and “clinical care pathways”. Keywords were then refined based on the relevance of the results, and additional terms were searched to survey related areas including “cardiometabolic risk factors”, ‘sarcopenia”, “exercise”, “body mass index (BMI)”, and “appetite”. We limited our search to studies from January 1995 onwards, with further relevant studies identified from citations within papers. Articles were initially screened for relevance based on the content of their abstracts. We focused primarily on clinical trials conducted in humans and also included in vitro and animal studies for mechanistic and molecular insights.

## 3. Obesity: Magnitude of the Problem

### 3.1. A Chronic Metabolic Condition

Obesity is traditionally defined as an excess of body fat, and is classically categorized in clinical practice in terms of body mass index (BMI). A BMI (in kg/m^2^) in the range of 18.5–24.9 is considered normal, 25–29.9 is overweight, and ≥30 is considered obese. Severe obesity, which is defined as a BMI over 40 kg/m^2^, is an alarming public health issue [6]. It should be noted, however, that BMI, although easy to gauge, has limitations when utilized as a diagnostic tool because it does not account for the exact muscle mass or fat mass, especially visceral adipose tissue. Visceral adipose tissue is more metabolically active and associated with the pathophysiology seen in obesity such as insulin resistance [7]. Waist circumference is an alternative to BMI and can be used as a complement to BMI. Kim et al. found a linear association between waist circumference and all-cause mortality in a study on 23,263,878 subjects over the age of 20 years [8]. Another reason fixation on BMI is not entirely accurate is because it does not account for factors such as age, sex, and ethnic variation. BMI also does not correlate with the risk of death at a population level [9]. Therefore, although BMI is appreciated for its simplicity in categorizing subjects, it should be used with caution as a diagnostic tool and might be better used as a screening aid [10].

Obesity is a non-communicable pandemic that continues to expand and affect over 1 billion people globally [11,12,13,14]. This worldwide escalation in population BMI is associated with multiple comorbidities and is believed to have numerous causes, including dietary habits, lifestyle, environmental stimuli, genetic predisposition, endocrinological disruptions, and gut dysbiosis [15,16].

### 3.2. Health Complications of Obesity

Obesity-driven inflammatory processes are responsible for a large portion of the damage inflicted by excess weight (Figure 1). Obesity has harmful effects on various body systems, most notably on the cardiovascular and endocrine systems, but also on the kidneys, liver, lungs, joints, and immune system [17]. The cardiometabolic consequences of obesity such as insulin resistance, glucose intolerance, type 2 diabetes, arterial hypertension, atherosclerosis, and dyslipidemia are all stressors on the heart and vascular system [18,19]. The lipid profile in obesity is marked by an increase in triglycerides and free fatty acids [20]. Obesity leads to oxidative stress from proinflammatory cytokines and adipokines [21]. The inflammatory environment incites endothelial dysfunction further contributing to cardiovascular risk and hypertension [22,23,24].


Figure 1Consequences of obesity-induced inflammation. Obesity leads to a physiological inflammatory response, such as oxidative stress, as illustrated by the hypertrophic adipocytes secreting pro-inflammatory cytokines, cardiometabolic conditions, and MALSD. These conditions increase the risk of mortality long-term which is generally associated with obesity.
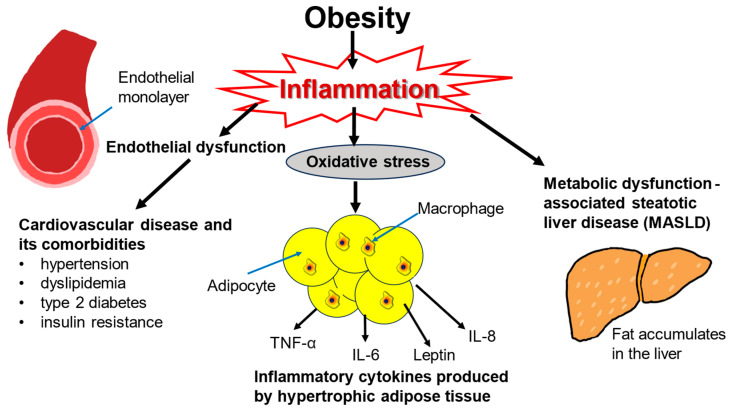



Obesity is also strongly associated with metabolic dysfunction-associated steatotic liver disease (MASLD) and metabolic dysfunction-associated steatohepatitis, which is inflammation-driven [25,26,27]. Obesity also elevates the risk of developing some types of cancer including colorectal, esophageal, liver, and kidney malignancies [28,29,30,31,32]. Both cardiovascular disease and cancer contribute to excess mortality associated with obesity [33,34]. Obesity is the strongest risk factor for obstructive sleep apnea [35]. In this condition, excessive adiposity may restrict the lungs and impede their inflation, and adipose tissue in the diaphragm may compromise muscle strength, leading to intermittent hypoxia and hypercapnia [36,37]. Other health issues associated with obesity include depression, anxiety, and chronic kidney disease [38,39,40].

## 4. Treating Obesity: Current Therapies

### 4.1. Dietary Approaches

Treating obesity has evolved in recent years, with various shifts in dietary, pharmacological, and surgical strategies available for the management of obesity [41]. There are various dietary styles with goals of controlling macronutrient composition and calorie restriction. For instance, high protein diet, Mediterranean-style diet, low carbohydrate diet, low-calorie diet, and low-fat diet are just several options [42,43].

Dietary advice historically has seen energy restriction as the foundation for weight loss. Numerous studies have demonstrated that dietary macronutrient composition is not the most significant contributing factor for weight loss [44,45,46]. However, guidelines have started to include a more nuanced understanding that there is a synergy between dietary nutrients and their food sources [47]. When looking at macronutrient content, initially, dietary fat, carbohydrate, and protein content are scrutinized. The low-or very-low fat intake approach is recommended for inducing significant short-term weight loss, but its long-term efficacy is not superior to dietary interventions with higher fat content [48]. The low carbohydrate diet involves consuming a low content of carbohydrates and a high content of fat and protein. It is especially suitable for individuals with type 2 diabetes and/or insulin resistance [49]. Exceeding 6-12 months of use may have undesired effects by increasing LDL cholesterol and cardiovascular risk in some studies, but others have found no difference [50,51,52,53]. In a randomized study of overweight adults, following a low-fat or high-fat diet with average or high protein resulted in an average 7% weight loss at 6 months, irrespective of diet type [45]. Regular physical activity is also recommended as a component of weight loss programs not only for energy expenditure, but for cardiometabolic health as well [54]. At this time, the evolution of dietary guidelines from isolated macronutrients to broader dietary patterns is receiving major attention. One such example is the Mediterranean diet, which is rich in plant-based food with high dietary fiber and antioxidants. The Mediterranean diet has a higher composition of fatty acids, unlike the conventional Western diet. In addition to its effect on body weight, a variety of health benefits have been ascribed to it. These include favorable effects on heart and brain health and decreased diabetes risk [55,56]. Much of the recommended annual weight loss diets often seen in US News and World Report reflect a variety of these dietary patterns, highlighting that diets are more than their nutrient content [57]. Finally, the microbiome has a variety of mechanisms through which it affects obesity, and pre/probiotic therapies could be a helpful addition to a weight loss regimen [58,59].

### 4.2. Non-Incretin Oral Pharmacotherapies

Pharmacological treatments for weight loss have expanded and, while GLP-1 agonists are the focus of this review, other choices are available and summarized here.

Perhaps the least successful of the FDA-approved weight loss drugs in terms of achieving weight loss maintenance is Orlistat (tetrahydrolipstatin). An older drug, orlistat inhibits pancreatic lipases which break down dietary fat. By reducing absorption of fat via the intestine, it promotes weight loss [60]. Patients are advised to consume a low-fat diet to combat the side effects of oily stool [61]. This drug leads to a weight loss nadir at around 36 weeks with a weight regain that happens at around 52 weeks [62]. However, the weight regain is relatively mild and by 104 weeks there is still overall weight loss. Anti-obesity medications such as Contrave (bupropion/naltrexone) or Qsymia (phentermine/topiramate) also appear to be effective for maintaining weight loss [63]. Admittedly, the lesser potency of these drugs in the initial weight loss phase often overshadows their potential for usage for weight loss maintenance purposes. For instance, the Contrave Obesity Research studies (COR-I, COR II, COR-BMOD, and COR diabetes) were performed over a 56-week period [64,65]. Contrave appears to modulate appetite and reward centers. For Contrave, a weight loss plateau seems to occur for all the COR studies around 32 to 36 weeks with overall weight loss around 8 to 9%. Qsymia has the longest-term data of the available oral anti-obesity drugs, upwards of 108 weeks [66]. This drug also appears to induce satiety and reduce hunger. Qsymia has demonstrated overall efficacy for weight loss maintenance, achieving sustained weight loss of 9-10% at the 108-week mark compared to 1.8% for placebo [67].

### 4.3. Endoscopic and Bariatric Interventions

The intragastric balloon is an anti-obesity intervention in which a silicone balloon is endoscopically deployed and filled with saline and inflated for 6 months. It is a temporary and minimally invasive therapy that reduces stomach capacity and results in decreased hunger and food intake [68,69]. It can be used as a primary treatment for obesity, as an alternative for patients who do not qualify for bariatric surgery, or as a bridge to surgery [70].

Bariatric surgery is indicated in patients with a BMI above 40 independent of coexisting comorbidities or in patients with a BMI over 35 with a history of comorbidities such as type 2 diabetes or hypertension [71]. There are two common procedures currently used: sleeve gastrectomy and gastric bypass. Emerging evidence suggests that the sleeve procedure is associated with few reoperations, but significant regain of weight while the bypass procedure can lead to more durable weight loss and glycemic control [72,73]. There is strong evidence to support that bariatric surgery results in greater long-term weight loss than even the top nonsurgical interventions and a recent retrospective study showed superior cardiovascular benefit for sleeve gastrectomy and gastric bypass compared to GLP-1 treatment over a 10-year follow-up period [74]. Variations of Roux-en-Y gastric bypass limb lengths have shown potentially increased weight loss and metabolic benefit, but also, possible early and late significant complications [75,76,77]. Possible complications of the gastric bypass include bowel obstruction and malabsorption, while possible complications of the sleeve gastrectomy include venous thromboembolism and gastroesophageal reflux disease [78,79].

The metabolic efficacy of bariatric surgery in increasing gut production of GLP-1 to supraphysiologic levels postprandially is considered a major factor in early weight loss [80]. GLP-1 agonists are being explored as an adjunctive therapy to combine with bariatric surgery to avoid the weight regain that can occur post-surgery [81,82].

This summary of current treatments for obesity highlights both its difficulty and importance, as well as obesity’s role in exacerbating other disease processes.

## 5. GLP-1 Drug Function and Mechanism

### 5.1. GLP-1—The Hormone

GLP-1 is a hormone released from the enteroendocrine cells of the small bowel in response to the arrival of a nutrient bolus [83]. GLP-1 is secreted continuously at low basal levels and rises within minutes of food ingestion [84]. The initial attention in the clinical space for GLP-1 was related to the glucose-dependent insulin secretion effect, often referred to as the incretin effect [85]. The additional properties of inhibition of glucagon secretion and inhibition of caloric intake accelerated the development of GLP-1 receptor agonists for usage in type 2 diabetes management [86]. While largely successful as an anti-diabetic drug therapy, the effects on both reducing food intake and promoting weight loss in persons with diabetes and animal models prompted further study as an anti-obesity medication [87,88].

### 5.2. GLP-1 and Energy Balance

Identifying the mechanism of action of GLP-1 receptor agonists for inducing weight loss naturally led to the study of the impact of this drug class on two key regulatory factors in weight regulation, namely energy intake and energy expenditure. Energy balance is dependent on nutrient intake and subsequent nutrient oxidation rates [89,90]. The energy-dense nature of fat makes it an efficient means of storing excess energy intake and thus the body favors fat for keeping energy in reserve [91,92]. Energy balance is buffered by fat stores and the adipose compartment therefore potentially producing an obesogenic state [93]. GLP-1 secretion seems to be impaired in obese subjects, which informs at least the partial role of GLP-1 in the pathophysiology of obesity [94,95,96].

There are conflicting data in animal models regarding GLP-1-related drugs stimulating energy expenditure [97]. The mechanisms behind this usually involved brown adipose tissue thermogenesis [98,99]. However, this has not translated over to human studies [100,101,102]. Therefore, most of the effect of weight loss via GLP-1-related pathways may be related to a decrease in energy intake, rather than the direct effects on energy expenditure [103]. Newer anti-obesity medications are being developed that have promise for altering energy expenditure, although these are multi-receptor agonists involving binding both GLP-1 receptors and receptors such as glucagon which has been associated with increased energy expenditure [104]. However, challenges remain in such multi-agonist receptor treatments, and the focus remains predominantly on energy intake [105]. For this reason, the role of GLP-1 within the nervous system continues to be thoroughly investigated.

### 5.3. GLP-1 and the Nervous System

GLP-1 receptors have been identified in numerous areas of the central nervous system (CNS) including the area postrema, hypothalamus, lateral parabrachial nucleus, nucleus accumbens, nucleus tract solitarius, and the vagal efferent neurons [106,107,108,109]. While the brain does produce GLP-1, most circulating GLP-1 is gut-derived [110,111]. There are often low levels of GLP-1 circulating systemically and the post-prandial rise in GLP-1 is of intestinal origin. However, in a murine model, knockout of glucagon genes originating from the bowel elicited no increase in food intake, and thus appetite suppression is attributed to GLP-1 produced within the CNS [112].

It is interesting to note that proglucagon and proglucagon-derived peptides (e.g., GLP-1, glucagon-like peptide-2, oxyntomodulin) are present in a small group of neurons in the nucleus tract solitarius within the brainstem and are another source of endogenous GLP-1 production [113,114]. The peripheral GLP-1 system and the central preproglucagon neurons within the brain stem have not been linked and appear to function independently of each other [106,115].

There is an existing network of neurons within the hypothalamus that is widely studied for its role in energy homeostasis. This is referred to as the melanocortin system and is composed of two differing populations of neurons involved in satiety and food intake. This includes the proopiomelanocortin and amphetamine-regulated transcript (POMC/CART) pathway, which promotes satiety and decreased hunger, often referred to as the anorectic pathway [116]. The opposing pathway is the agouti-related peptide (AgRP) neurons which stimulate hunger and increase food intake, often referred to as the orexigenic pathway [117]. In rodent models, the majority of POMC/CART neurons have GLP-1 receptors, predominantly in the arcuate nucleus of the hypothalamus [118]. However, GLP-1 activity has also been seen in the hindbrain, with infusion of GLP-1 analogs into the fourth ventricle in rodent models. Reductions in food intake and body weight have been found, implying that both the hypothalamus and brainstem are important in the control of energy intake and body weight [119,120]. If these findings are also reflected in human physiology, targeting GLP-1 within the CNS would be a constructive goal for pharmacologic strategies.

### 5.4. GLP-1 Benefits to Organ Systems

It is meaningful to note that GLP-1 agonists have been increasingly examined outside the effects on type 2 diabetes and obesity. A growing number of trials have studied the beneficial impact of GLP-1 agonists on metabolic dysfunction associated with steatotic liver disease (MASLD), chronic kidney disease, as well as cardiovascular disease [121,122,123,124]. The benefit of the GLP-1 agonists may be from the reduction of adiposity, or non-adiposity related. For instance, the SELECT trial demonstrated the cardiovascular benefit of the GLP-1 receptor agonist semaglutide beyond that of weight loss [125]. In this placebo-controlled trial, subjects were over the age of 45 years, had a BMI of 27 or above, and established cardiovascular disease. Cardiovascular endpoints (death from cardiovascular causes, nonfatal myocardial infarction, or nonfatal stroke) were compared in those taking semaglutide versus placebo after a mean follow-up of 39.8 months, and the drug intervention was found to be superior to placebo. Benefits to the cardiovascular system extend beyond weight loss to affect other risk factors such as triglyceride level, systolic blood pressure, risk of progression to diabetes, and the inflammatory marker C-reactive protein. SELECT also showed better cardiovascular outcomes in persons with obesity and without diabetes who had previously undergone coronary artery bypass graft surgery [126]. Weight loss was sustained over 4 years [127]. There are a number of proposed mechanisms including reduction of inflammation, improvement of endothelial and ventricular function, as well as decreasing platelet aggregation [128]. A meta-analysis from Lin et al. showed the benefit of GLP-1 drugs in peripheral artery disease and heart failure [129]. Zhang et al. found that in persons with type 2 diabetes, GLP-1 drugs offered benefits whether or not the patients were also taking metformin [130].

## 6. GLP-1 Receptor Agonists and Weight Loss

Liraglutide was the first GLP-1 receptor agonist that was FDA-approved for weight loss. It helped one-third of the non-diabetic study patients achieve a loss of 10% of their body weight and also helped them sustain their weight loss for upwards of 1 year [131]. The weight loss plateau occurred after about 20 weeks and continued until the drug was stopped at 56 weeks. Similarly, the subcutaneously once weekly formulation of the GLP-1 receptor agonist semaglutide showed equally promising results for weight maintenance [132]. First, there was a greater overall weight loss of 15% and therefore the weight loss plateau was delayed to around 68 weeks. Persistence of the weight loss plateau (or presumed weight loss maintenance) occurred up until 104 weeks [133].

The newest incretin-based medication is a combined GLP-1 receptor and glucose-dependent insulinotropic polypeptide (GIP) dual agonist known as tirzepatide. An even greater weight loss is seen with this novel dual agonist, achieving upwards of a 22.5% weight loss at 72 weeks [134,135,136]. At 3 years of follow-up, tirzepatide use led to a sustained mean loss of weight of 20% with less likelihood of deterioration to diabetes in persons with obesity and prediabetes when compared to placebo [137].

## 7. GLP-1 Usage and Adverse Effects

### 7.1. Usage

Since the FDA approval of liraglutide for weight loss in 2015, the use of this class of medication has exploded, particularly over the last few years with the availability of weekly GLP-1s including tirzepatide and semaglutide. In an analysis of one US health systems database a 700% increase in GLP-1 prescribing over the past four years was noted, primarily driven by prescriptions for obesity [138]. Such a rapid increase in usage led to multiple drug shortages beginning in 2022 which continued through late 2024 [139]. Under section Section 503A of the FD&C Act compounding pharmacies were permitted to produce GLP-1 compounds to increase availability [140]. The usage of compounding pharmacies does come with risks as the medications produced are not FDA-regulated [141]. Furthermore, ongoing surveillance and monitoring led to the FDA issuing a statement of counterfeit products in users’ hands [142]. It is important to note when discussing the side effects and adverse outcomes of this drug class that many adverse effects are class-related; however, the degree and severity of reported side effects in certain circumstances may have been due to drug purity or dosing issues.

### 7.2. GLP1 Receptor Analogs Approved for Weight Loss

The potential application of GLP-1 in diabetes was recognized as early as 1998 [143]. The emergence of GLP-1 receptor agonists has re-invigorated interest in anti-obesity medications and more effective weight management. GLP-1 receptor agonist drug development began in earnest in 2005 with the approval of exenatide, a synthetic form of a natural peptide hormone isolated from the saliva of the venomous lizard Gila monster [144]. It was found to have similar activity to GLP-1, but with a longer half-life. A synthetic version of exendin-4 was approved by the FDA in 2005 for glycemic control in type 2 diabetics [145,146]. A number of clinical trials in persons with diabetes have been subsequently performed and frequently cited for the clinical efficacy of exendin-4 [147,148]. However, the prevalence of obesity along with the need for more effective anti-obesity treatments prompted the study of the first daily GLP-1 receptor agonist, liraglutide. Liraglutide was the first injectable daily GLP-1 receptor agonist that was approved by the FDA for weight loss in 2014. The Satiety and Clinical Adiposity Liraglutide Evidence (SCALE) trial demonstrated one-third of patients lost 10% of their body weight in 1 year with sustained weight loss demonstrated at 2 years [149,150]. A 3-year extension of the SCALE trial showed that persons with overweight or obese and prediabetes taking liraglutide had a reduced risk for developing type 2 diabetes with greater weight loss compared to those taking a placebo [151]

Arguably, the arrival of semaglutide more than five years later sparked the public’s attention to GLP-1 agonist therapy. This once-weekly subcutaneous injection was FDA-approved for weight loss in June 2021. Semaglutide was studied in a comprehensive series of clinical trials known as the Semaglutide Treatment Effect in People with Obesity (STEP). The STEP-1 trial is considered the pivotal trial that demonstrated 14.9% weight loss at 68 weeks with semaglutide 2.4 mg [142]. The follow-up STEP-5 trial demonstrated that semaglutide could sustain weight loss over 104 weeks or nearly 2 years [152]. While there are newer and even more potent anti-obesity drugs on the horizon, semaglutide still garners attention for its non-diabetic and non-obesity benefits, specifically cardiovascular risk reduction, as well as improving renal outcomes [125,153].

However, tirzepatide is a novel dual agonist drug that activates GLP-1 receptors, as well as the GIP receptor. It was FDA-approved for weight loss in November 2023 [154]. Tirzepatide can uniquely induce weight loss beyond what is achieved with selective GLP-1 agonists alone. As with the STEP trials, a comprehensive series of clinical trials were performed with tirzepatide known as the SURMOUNT trials. The SURMOUNT-1 trial demonstrated a 15 mg dosage of tirzepatide in non-diabetic obese subjects leads to 20.9% weight loss at week 72 with sustained weight loss during a 3-year extension period [134,137].

### 7.3. Common Adverse Effects

GLP-1 medications can cause a range of side effects related to the gastrointestinal system as well as changes in muscle mass and effects on the appearance of the face and loss of hair (Figure 2). The most common detrimental consequences of the GLP-1 class are gastrointestinal. The most often reported side effects are nausea and vomiting which are a result of activation of specific GLP-1 receptors in the hindbrain and these symptoms can be mitigated with gradual dose escalation [155,156,157]. In addition to nausea and vomiting, other GI-related side effects include diarrhea, constipation, dyspepsia, decreased appetite, and abdominal pain [158,159,160]. It is estimated that around 80–90% of patients will develop an adverse effect from the use of this class.

A more serious gastrointestinal concern is that of pancreatitis. Early studies on patients with type 2 diabetes treated with incretin therapy including GLP-1s and dipeptidyl peptidase-4 (DPP4) inhibitors did demonstrate an association between drug usage and the development of pancreatitis [159]. GLP-1R analogs have been associated with lipase and amylase suggesting a mechanism of pancreatic inflammation [161,162]. In fact, the package insert for semaglutide suggests an increase of 13% for amylase and 22% for lipase and the dual incretin agonist tirzepatide suggests a 38% increase and lipase upwards of a 42% increase [163]. Despite this, animal studies have demonstrated decreases in pancreatic secretion in response to GLP-1 elevation, therefore the mechanism behind this potential interaction of GLP-1 receptor analogs and pancreatitis remains elusive [164]. It is important to note that these studies mostly included dipeptidyl peptidase-4 DPP4 and early GLP-1s, exenatide, and liraglutide. Furthermore, patients with type 2 diabetes are inherently at higher risk of pancreatitis [165]. The large and long-duration GLP-1 cardiovascular outcome trials did not show an increase in pancreatitis [166]. Recent trials on the newer GLP-1 agents including tirzepatide and semaglutide specific to weight loss have not demonstrated increased rates of pancreatitis, although product labeling still instructs avoidance in patients with a history of pancreatitis [167,168,169].

Rapid weight loss has long been associated with biliary disease, sarcopenia, and alopecia [170,171,172,173,174]. As the newer agents in the GLP class have become incredibly potent where users are losing an estimated 15-20% of body weight, with much of the weight loss occurring in the initial weeks of initiating the drug [175,176]. It is thus not surprising that these same side effects of rapid weight loss are seen as a class effect. The rapid weight loss can be visualized in many areas of the body and one of these manifestations known as “Ozempic face” occurs when fat pads in the face are rapidly depleted [177,178]. Patients should be aware of these potential unwanted effects and, to minimize loss of muscle mass, encouraged to participate in resistance exercises and increase protein intake [179].

While these cosmetic findings are an issue, the loss of lean body mass is another area of concern [180]. The landmark GLP-1 drug trial for semaglutide, STEP 1 (semaglutide treatment effect in people with obesity), demonstrates a significant loss of total lean body mass [132], which has been further corroborated by other investigators [181]. Tirzepatide has demonstrated total lean mass loss as well, although additional studies are needed to determine the impact of this [134,182]. The question of whether the ratio between fat mass and lean body mass is disrupted or maintained during weight loss with GLP-1 agonists is still unresolved [183,184]. Studies to clarify this issue would particularly be needed for patients who are afflicted with sarcopenic obesity, a condition of a mismatch between muscle and fat mass.

### 7.4. Less Recognized Adverse Effects

A common finding amongst all GLP-1 trials was that of increased heart rate attributed to a direct effect on the pacemaker cells within the sinus node of the heart [185]. This is generally a benign clinical finding with no associated arrhythmias from GLP-1 usage [186,187]. Headaches, dizziness, hypoglycemia, and injection site reactions have also been reported albeit to a much lesser degree [188,189]. Xie et al. applied a high dimensional approach to analyze associations between GLP-1 treatment and health outcomes in over 1.9 million persons over an average of 3.68 years based on data from the US Department of Veterans Affairs and found, in addition to gastrointestinal and other common symptoms, a significant increase in hypotension, joint pain, kidney stones, and nephrolithiasis [190].

### 7.5. Special Considerations

#### 7.5.1. Ocular

GLP-1 therapy has two ophthalmological manifestations warranting discussion [191]. Ever since the UK Prospective Diabetes Study demonstrated modifiable retinopathy with improvements in glycemic control, clinicians and patients have aimed to improve glucose as a standard of management in type 2 diabetes [192]. A somewhat paradoxical effect has been demonstrated with GLP-1 usage and other agents for type 2 diabetes in which rapid improvement in glycemia results in worsening of retinopathy [193]. The long-term effect of GLP-1 on retinopathy in patients with type 2 diabetes may in fact be beneficial. It is important to monitor patients on therapy [194,195,196].

More recently with the increased employment of GLP-1 agents, a relatively new complication, non-arteritic anterior ischemic optic neuropathy (NAION), has been associated with their usage. Although both the association of retinopathy and NAION can be seen with GLP-1 use, it is worth noting that a majority of cases occurred in patients with type 2 diabetes. It is unclear whether those without type 2 diabetes using GLP-1 for weight loss are at the same risk [197,198].

#### 7.5.2. Malignancy

Parks and Rosebrough first expressed concerns regarding human safety with liraglutide as early rodent trials demonstrated an increased risk of medullary thyroid carcinoma [199]. All GLP-1 agents have carried an FDA black-boxed warning of increased risk of C cell thyroid carcinoma and recommended agents used in patients with a personal or family history of multiple endocrine neoplasia type 2A or 2B. The exact risk change with other histological subtypes of thyroid carcinoma is yet to be fully established. Two trials have supported an increase in the risk of all types of thyroid carcinoma [200,201]. In contrast, a more recent Scandinavian trial did not detect a significant association between GLP-1 usage and thyroid cancer [202]. All studies are in agreement that the greatest risk if any does occur in the initial months or year of therapy [200,201,202,203,204]. Emerging data have established that GLP-1 agonist administration has not increased the risk of malignancy outside of the thyroid gland and may in fact reduce the risk of malignancy with potential for preventive applications [203,204].

#### 7.5.3. Pregnancy

In animals exposed to GLP-1 agonists during pregnancy or lactation, studies have shown an association with reduced offspring size, delayed growth, and skeletal deformations, each of which is typically associated with reduced maternal caloric intake. Accidental human evidence has not been associated with any adverse outcomes, but overall data are scarce [205,206]. Interestingly enough, pregnancy rates may actually increase while on GLP-1 medications. Firstly, with delayed gastric emptying, nausea, vomiting, and diarrhea there is a potential for impaired absorption of oral contraceptive pills. Specific guidance on the usage of alternate methods of birth control is provided by drug manufacturers. Secondarily medically induced weight loss, particularly when totaling in excess of 5% of total body weight has demonstrated effectiveness in improving fertility [207].

#### 7.5.4. Mental Health

Initial case reports and media reportage of an increase in suicidal ideation in persons taking GLP-1 medications were looked into further by the FDA. The FDA and studies from a cohort of Scandinavian patients concluded no association between GLP-1 use and suicidal ideation, self -harm, or new onset of depression [208,209,210]. Data are still accumulating and no definitive answer has emerged [211,212].

#### 7.5.5. Perioperative

Early case reports called into question appropriate fasting times for pre-procedural and operative fasting due to retained gastric contents and risk of aspiration in patients taking GLP-1 medications and compounds [213,214]. A recent multi-society joint guidance statement advocated for an individualized approach based upon each patient’s unique factors rather than a one-size approach of holding this medication for all patients undergoing procedures [215]. Though more evidence is needed such guidance is useful to patients and clinicians at the present time [216,217]. As with all new medications or those whose use increases due to expanded indication, ongoing monitoring and close surveillance by both patients and clinicians continue to be necessary.

In summary, there are many side effects associated with GLP-1 drug treatment and they span a range from minor to potentially life-threatening. For many of them, measures can be taken to alleviate discomfort and risk (Table 1).

## 8. GLP-1 Receptor Agonists and Weight Regain

### 8.1. Forces Driving Weight Regain

The general trajectory of weight loss with initiation of GLP-1 therapy has been well-studied (Table 2) [218,219,220]. Despite the overall efficacy of the incretin-based treatments for weight loss, there is a lack of long-term controlled studies beyond about 4 years available [221,222]. While there is optimism that continuing use of GLP-1 treatments will preserve weight loss, most other anti-obesity strategies, including surgical interventions, generally have weight recidivism [127]. This can be attributed to the persistent effects of metabolic adaptation, the phenomena seen in weight regulation that may cause weight regain and potentially a weight loss plateau [223]. Definitions of weight regain may vary, namely the duration and how much is considered significant. It is worthwhile to point out that a weight loss plateau is often stated with the assumption that additional weight loss is desired, but difficult to achieve. Additionally, there are those who are attempting to prevent weight regain after already achieving a weight-reduced state. In this setting, prevention of weight regain may be better-termed weight loss maintenance.

Ultimately, understanding the existing forces that occur in a weight-reduced state may help to understand what may drive weight regain [224,225]. The conceptual framework for weight regain and weight loss maintenance is based on the theory that the human body acts to defend a particular body mass, via the hypothetical “settling point” of weight [226,227,228]. The drivers for weight regain are hypometabolism and hyperphagia in the weight-reduced state [229,230,231]. Hypometabolism, or the decrease of energy expenditure greater than what would be predicted, is known as metabolic adaptation [232]. It is therefore rational to consider these processes as targets to prevent weight regain and achieve weight loss maintenance, although some studies show that energy expenditure is not uniformly disproportionately decreased in those sustaining weight loss long-term [233]. While there are available therapies for hyperphagia and hunger and, in fact, appetite reduction is a key effect of GLP-1 agonists, there are no significant available therapies that can address the decrease in energy expenditure [234]. However, if one were to choose a mechanism to prevent weight regain, targeting hunger would seem to be the better alternative due to the greater effect of metabolic adaptation on hunger, rather than the decrease in energy expenditure [235].

**Table 2 biomolecules-15-00408-t002:** GLP-1 agonist intervention and observed effects over time.

Duration of Intervention with GLP-1 Agonist (in Months)	Observed Beneficial Effects	References
BaselineStarting dose	Body adjustment, appetite reduction, decreased caloric intake	[234]
1-3 monthsStep-up dose	Weight loss, improvement in insulin sensitivity, reduction in HbA1c	[121,122,158]
3-6 monthsStep-up dose	Continued improvement in blood sugar levels, most effective weight loss, reduced risk for cardiovascular events	[121,158,224,225]
6-12 monthsStable dose	Weight loss plateau, improvement in HbA1c, further decrease in cardiovascular events	[123,158]
Beyond 12 monthsStable dose	Well-controlled HbA1c, prevention of long-term diabetes complications, weight loss maintained	[123,124,132,158]

Abbreviations: GLP-1—glucagon-like peptide-1; HbA1c—hemoglobin A1c.

### 8.2. Hunger

For those with untreated obesity and seeking active weight loss, decreasing hunger and achieving caloric restriction is seemingly the primary process that needs to occur. Therefore, treating hyperphagia is the strategy for both weight loss and weight loss maintenance. However, even during active weight loss, metabolic adaptation seems to be set into motion, and in fact may be triggered by achieving 11% of total body weight loss [236,237]. This suggests that decreased caloric intake can drive the initial periods of weight loss, but when weight reduction reaches 10%, decreases in energy expenditure begin with increasing strong driving forces of hunger to try to slow the weight loss process. Developing treatment paradigms for weight loss maintenance remain focused on decreasing hunger, despite the compensatory decrease in energy expenditure [94]. Incretin-based medications and other anti-obesity medicines target hunger, therefore fostering both weight loss and weight loss maintenance.

The popularity of GLP-1 receptor analogs may have to do with their profound effects on the CNS. Early studies demonstrated the anorectic actions of GLP-1 on the hypothalamus [238,239]. In situ hybridization studies in animal models demonstrated GLP-1 receptor presence in many other brain areas such as the thalamus, nucleus accumbens, and hindbrain. GLP-1 action in the hindbrain reduced food intake and body weight over time [240]. Clinical studies have supported this with GLP-1 receptor analogs such as liraglutide and semaglutide in both weight loss and weight loss maintenance trials.

### 8.3. GLP-1 Receptor Agonist Discontinuation

Cessation of these drugs to see if weight maintenance could be achieved was largely unsuccessful [241] (Table 3). Randomized double-blinded placebo-controlled withdrawal studies were performed in both semaglutide and tirzepatide with crossover to placebo at 20 weeks and 36 weeks, respectively [242,243]. When switched to placebo there was invariably weight regain, implying that the loss of inhibition of hyperphagia drove this process. It is worthwhile to note in both studies all participants were prescribed a reduced calorie (500 kcal/day deficit) and increased physical activity 150 min/week) regimen, which was insufficient to help preserve the initial weight loss. It is interesting to note that an extension study of semaglutide was performed to 120 weeks, but treatment was discontinued at 68 weeks. As expected, there was weight regain, but there still was an overall 5.6% net loss of weight by the end of 120 weeks [244]. The authors of this study pointed out that there appears to have been a slowing of weight regain towards the end of the study, implying a weight loss plateau below the initial pre-treatment weight. This could imply the drug’s potential altering of the settling point for weight.

## 9. Avoiding Weight Regain

### 9.1. Factors That Avert Weight Regain

A large public database known as the National Weight Control Registry (NWCR) was the first study to identify participants who were successful at weight loss and follow them over a 10-year period, with an attempt to identify variables that were associated with success [245]. To participate in the study, weight loss greater than 30 pounds had to have been maintained for more than 1 year at the time of enrollment. In this study population, 88% were able to keep 10% of their body weight off at year 5 and 87% at year 10. These successful subjects with weight loss maintenance reported high levels of physical activity, high levels of dietary restraint, low calorie, and fat intake, and low levels of overeating (loss of control of eating or disinhibition) [246]. Data from the NWCR database highlight the role of negative thoughts that induce eating. It is worthwhile to note these variables are all related to appetite, hunger, and caloric intake. The NWCR continues to provide information about these successful weight loss maintenance patients, but analyzing this data should be taken with the knowledge that these are select, highly motivated subjects who are attentive to their own health needs. Other studies confirm the importance of dietary restraint and physical activity in preventing weight regain [247,248].

Even more recently in 2022, a symposium was convened to discuss the state of the science of weight loss maintenance, known as the Pennington Biomedical Scientific Symposium [249]. The statement generated by the symposium broadly included nutritional strategies and physical activity recommendations. Not surprisingly, food composition is often an area of question by both scientific communities and the food industry to determine the right “mix” of macronutrients to facilitate weight loss and weight loss maintenance. However, over the years, many studies have shown no differences in achieving weight loss from a variety of macronutrient approaches [45] and calorie reduction is likely the more successful approach for weight loss maintenance [250,251]. However, as detailed in this review, calorie reduction generally leads to short-term weight loss, with poor success rates for long-term weight loss maintenance. This can be attributed to the metabolic adaptations that are seen to occur in those with obesity. Most effective weight loss from a dietary standpoint is seen over 3 to 6 months, with at least one-third of patients regaining lost weight within the first year and the majority of patients regaining the weight after five years [224,225].

The Pennington symposium highlighted potential alternative approaches for nutrition management that may be beneficial for weight loss maintenance. Specifically, a decrease in processed and ultraprocessed food consumption would be beneficial for weight loss maintenance. Processed foods are difficult to remove from the population’s food supply and public diets, due to their ubiquity as well as their low cost, convenience, and appealing taste [252,253]. However, consumption of ultraprocessed food has been shown to induce an even greater consumption of calories, and therefore leads to weight gain. Alternatively, an upcoming area in nutrition sciences involves precision medicine. Precision medicine itself is an area of medical management that tries to match personalized treatments or food content, to individual genetics, microbiome, metabolism, age, and sex. However, this treatment strategy is still in the early stages although the National Institutes for Health (NIH) recently has invested in research in the area of personalized nutrition.

### 9.2. Level of Physical Activity

While most of the interventions are based on targeting hunger and energy expenditure, there is little understanding of whether physical activity or exercise plays a role [254]. In simple terms, exercise is related to energy expenditure, and therefore increasing exercise increases energy expenditure and therefore weight loss. While this does seem to be true over the lower ranges of physical activity, with the upper ranges of physical activity the energy expenditure appears to plateau, consistent with a “constrained total energy expenditure model” [255,256]. Despite its somewhat attenuated impact on energy expenditure, there is evidence that exercise still helps to achieve weight loss maintenance. For instance, those with high levels of physical activity are more successful at weight loss maintenance [257,258,259]. Other studies have found little to no impact of physical activity on maintaining weight loss [260]. On the other hand, Ostendorf et al. found that high levels of physical activity including 200-300 min/week of at least moderate-intensity aerobic activity for upwards of 18 months supported weight loss maintenance [261]. Data from the NWCR also suggest successful weight maintainers can spend upwards of one hour per day in light physical activity [262]. Newer strategies for weight maintenance have focused on preserving or even increasing lean body mass to counteract the decreases in energy expenditure thereby allowing for sustained weight loss [105,263].

### 9.3. Targeting Mood and Providing Support

Psychological well-being is an important aspect of both eating behaviors and weight loss [264,265]. Successful weight loss management is often associated with low levels of depression [266,267]. Studies suggest that GLP-1 drugs are not a direct cause of depressive symptoms in weight loss [268]. A supportive weight management team approach considers mood changes and how they can affect quality of life [269,270]. Family encouragement and support have also been associated with successful weight loss management [271,272,273]. For health care practitioners, in-person sessions may be more successful than remote sessions, but both have value [246,274,275].

## 10. Present and Future of Weight Loss

It may be possible to maintain weight loss while tapering GLP-1 to a lower dosage or prolonging the time between doses [276,277]. Patients have also been reported to take “drug holidays” in which they pause the use of the drug intermittently for special occasions, but there is very little in the literature on this [278]. The effects on weight, cardiovascular health, and other parameters of decreasing the dose or pausing and resuming the use of GLP-1 agonist is an area where evidence-based studies are needed.

The approval of tirzepatide, a novel long-acting dual incretin agonist of both GLP-1 and another incretin, GIP, continues to create excitement for the development of anti-obesity medications [279]. GIP is a similar gut hormone to GLP-1 that is secreted by enteroendocrine cells. It similarly induces insulin secretion in the setting of hyperglycemia. However, earlier studies showed inconsistences of GIP as a cause of weight loss, although more recent studies have demonstrated increased weight loss efficacy [280,281,282,283]. Recent studies have now demonstrated the strongest weight loss effect with the dual agonist for GLP-1 and GIP, upwards of 22% over 1 year [134]. The effect of tirzepatide on energy expenditure appears negligible, therefore making the effect on energy intake the most impactful [284]. However, available studies are only seen in rodent studies, but the additive or synergistic effects of GLP-1 and GIP on hunger and satiety require further clinical research [285].

A multi-agonist approach is a likely road for the future of anti-obesity drug development involving novel receptors such as glucagon and amylin possibly with even more profound weight loss [286,287]. Amylin and its analogs are gaining much interest [288]. Amylin is a non-incretin hormone produced in the pancreas by β cells that is released upon nutrient intake. It delays gastric emptying and acts on areas of the brain controlling appetite and signaling fullness. Amylin analogs such as cagrilintide are being explored for obesity treatment in concert with GLP-1 drugs [289].

Drugs are in development to combat muscle loss during fat loss [290]. Monoclonal antibodies such as bimagrumab, trevogrumab, and garetosmab, prevent muscle loss by blocking the activation of receptors within the myostatin-activin-follistatin-inhibin system that cause muscle mass to decrease [291]. Clinical trial outcomes will determine whether these will be useful either on their own or in conjunction with GLP-1 agonists.

GLP-1 agonists demonstrate efficacy for weight loss maintenance, but only while the patient is continuing to use the medication. The NCWR highlights variables associated with weight loss maintenance success for upwards of 10 years, although this study is with a motivated population of individuals [243,292]. Certainly, teaching patients to be mindful of their eating and to consume adequate protein can contribute to weight loss maintenance success and overall health [293]. Physical activity also has shown significant benefits for upwards of two years. What remains to be seen is if the mixing and matching of the initial weight loss strategy, whatever this may be, with another weight loss maintenance strategy will lead to successful weight maintenance. An example would include the usage of liraglutide that helped one achieve a particular amount of weight, but continued usage of the drug led to weight regain, would switching to Contrave help to achieve weight loss maintenance?

These are questions that will need to be answered for future clinical trials. Ideally, the chosen initial intervention for weight loss would also be effective for weight loss maintenance. However, the only truly long-term strategy that has been the most successful for long-term weight loss is surgical weight loss. Upwards of 20 years has seen a sustained 22% weight loss [294]. However, even surgical weight loss reaches a peak weight nadir 1 to 2 years after surgery and weight regain tends to occur after. However, the overall weight loss is still significant, and surgical weight loss is ultimately a personal choice for patients as it does carry risks [295,296,297].

## 11. A Multi-Pronged Approach to Avoiding Weight Regain

The use of GLP-1 drugs as a weight loss tool is prevalent and effective, but it is preferable to find ways to keep the weight off without a lifetime of drug treatment and this is an area that needs attention [298]. Lifestyle interventions alone are not durable for most people [299,300]. Studies are clearly needed to find ways to support withdrawal of pharmacotherapy without weight rebound using a multi-disciplinary approach which would likely involve behavioral change, nutritional guidance, structured physical activity, and perhaps peer support groups [301,302,303,304,305].

## 12. Conclusions

Even with information on nutrition, physical activity, anti-obesity medications, and psychological support, there is no universally effective strategy in terms of weight loss maintenance. The overall mechanisms of GLP-1 agonists on weight loss are predominantly through the reduction in energy intake and not on energy expenditure. While the area of anti-obesity medication development is expanding, GLP-1 receptor agonists are already available and represent substantial progress in the growing armamentarium for use in weight loss. The effect seems principally within the CNS, but combination treatment of GLP-1 with other targets appears to further improve weight loss in early clinical trials. At this time, very little attention is being afforded to the discontinuation of GLP-1 drugs after weight loss has occurred, but this may change if serious consequences of prolonged exposure in young persons are documented.

## Figures and Tables

**Figure 2 biomolecules-15-00408-f002:**
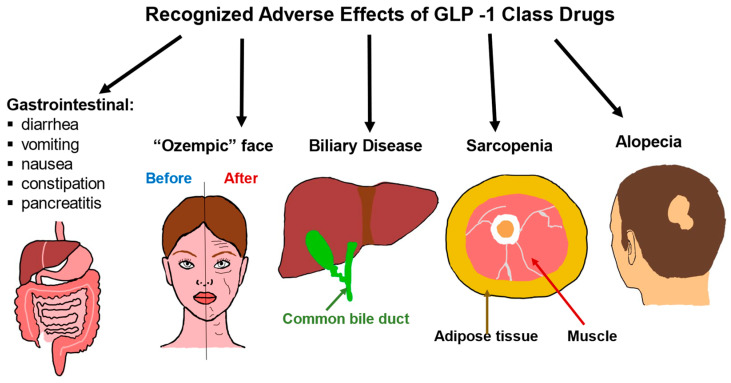
Adverse effects of GLP-1 class drugs. Gastrointestinal issues are common, ranging from nausea and diarrhea to rarer and more severe consequences such as pancreatitis. Rapid weight loss due to the use of GLP-1 drugs is associated with biliary disease, sarcopenia, and alopecia. Rapid weight loss can also lead to what is known as an “Ozempic face”, where the cheeks become hollowed out, and wrinkles, as well as eye bags, become more pronounced.

**Table 1 biomolecules-15-00408-t001:** Common adverse effects of GLP-1 agonists and approaches to minimizing these consequences.

Adverse Effect	Mitigating Strategies	References
Nausea, vomiting	Gradual increase in dose, small meals, anti-emetics	[156]
Diarrhea	Hydration, low fiber foods, reduce consumption of dairy, coffee, alcohol	[156]
Constipation	Encourage physical activity, hydration, ample fiber in diet	[156]
Pancreatitis	Discontinue drug, standard treatment for pancreatitis	[156]
Alopecia	Change to a different GLP-1 medication, topical hair loss treatments	[174]
“Ozempic face”	Cosmetic procedures such as facelift, dermatologic fillers, autologous fat transfer	[177,178]
Sarcopenia	Exercise (emphasize resistance-training), increase protein intake	[171,180]
Gastroparesis with anesthesia	Discontinue at least one week prior to procedure	[158,217]

**Table 3 biomolecules-15-00408-t003:** Studies of weight regain after GLP-1 agonist or dual GLP-1/GIP agonist discontinuation.

GLP-1 Agonist Used	Period of Agonist Treatment	Observed Weight Loss (%)	Weight Regain After Withdrawal (%)	References
Semaglutide	68 weeks	17.3	11.6	[244]
	68 weeks	7.9	6.9	[224]
Liraglutide	56 weeks	6.2 (after 6% loss on low-calorie diet alone)	1.9	[131]
Tirzepatide (GLP-1/GIP dual agonist)	36 weeks	20.9%	14	[243]

Abbreviations: GLP-1—glucagon-like peptide-1; GIP—glucose-dependent insulinotropic polypeptide.

## Data Availability

No new data were created or analyzed in this study. Data sharing is not applicable to this article.

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
