# Peer review of "Weight Reduction with GLP-1 Agonists and Paths for Discontinuation While Maintaining Weight Loss"

_biomolecules, 2025, doi:10.3390/biom15030408_

Round 1
Reviewer 1 Report
Comments and Suggestions for Authors
In this review, the authors discuss the health risks of obesity and the magnitude of the problem. They discuss diets, physical activity, bariatric surgery and pharmacologic therapies. The mechanisms of action of the GLP-1 receptor agonists are discussed as well as the clinical trials in weight management and their potential role in weight maintenance. They claim to focus on benefits, risks and unknown effects. They hypothesize that weight loss could be durable without pharmacotherapy.
Comments on the abstract as well as most of the review: There are very many reviews available with a focus on most of what is stated here in the abstract. But where are the “Paths for Discontinuation While Maintaining Weight Loss?”. If they have a good suggestion this might be brought up in the abstract already, because this is a subject oif great current interest. Unfortunately, the authors do not really provide concrete suggestions that could be followed by patients and their doctors. This is the greatest disappointment regarding the review.
Specific comments:
l. 52 “the lack of long-term clinical trials” and “unforeseen long term side effects” are of course correct, but on the other hand the GLP-1 Ras have been on the market since 2005, and some of the very large studies (CVOTs and others) extend for up to 4-5 years, which are quite remarkable durations. The authors disregard the information that has accumulated during the most recent years
l. 99: “(MASLD)…….is inflammation-driven” – not clear- it is driven by obesity, while MASH is inflammation
Fig.1 Is leptin “an inflammatory cytokine”?
Section 3.1 on Dietary approaches is superficial and contains nothing new
l. 163 The nice effects of Qsymia are not corrected for placebo
section 3.3 is also very superficial
l. 195 The effects on food-intake and weight loss are said to be self-evident? How is that? (But the decision to look at them for obesity therapy might seem self- evident) (although it was not, because of the side effects prevented people from trying higher doses and more powerful agonists had to be developed )
l. 222 “it is perhaps a common misconception that most GLP-1 secretion is gut derived.” If you are talking about circulating GLP-1 this is actually the case. The quantity of GLP1 produced in the periphery is also a lot larger. Actually this section need rewriting. The rise after bariatric surgery is certainly derived from the gut. It is important to note that no connection has been found between the peripheral GLP-1 system and the central (the ppg neurons of the brain stem).
L. 253: how did the “SELECT trial demonstrate cardiovascular benefits of semaglutide beyond the weight loss”?
l. 261. There was a three year extension of the SCALE 1 trial; also note that the SELECT trial showed a weight loss plateau of 4 years’ duration.
l. 272, there is also a three year’s extension of SURMOUNT 1 with tirzepatide.
l. 293: exenatide was the first GLP-1RA to be approved, but work on the GLP-1 agonist had begun long before that – the use of stabilized forms of GL-1 had begun already in 1998. It was the renal elimination that was the real hurdle (the iv. half-life of exenatide is only 30 min)
l. 326. The effects of the GLP-1RAs on nausea and vomiting are not related to the effects on gastric emptying but are due to interactions with brain receptors . It is true that side effects can be elicited in many patients if you overdose, but more importantly they can be mitigated by slow up-titration.
345 . Most important regarding the risk of pancreatitis is to mention the CVOTs where the risk was minimal or absent, not because “they are recent” but because they are large and of long duration
l. 355. Most important regarding the lean body mass loss is the ratio between fat mass and lean body mass. This ratio is generally maintained during GLP-1RA induced weight loss, indicating that there is no specific loss of lean body mass. Importantly, however, for some patient groups any loss can be a problem
372. The mechanism behind the increase in heart rate has now been elicited ( Cardiovasc Res. 2024 Oct 14;120(12):1427-1441)
Regarding the side effects of the GLP-1RAs the authors should describe the findings in the recent publication in Nature Med, https://doi.org/10.1038/s41591-024-03412-w
Author Response
We thank the reviewer for thoroughly scrutinizing our manuscript. As requested, we have revised the manuscript and addressed the specific comments of the reviewer. The revised sections are delineated in red in a marked copy of the manuscript text.
Below, we provide a point-by-point response to the reviewer’s comments.
Reviewer # 1 Comments and Responses
- COMMENT #1: But where are the “Paths for Discontinuation While Maintaining Weight Loss?”. If they have a good suggestion this might be brought up in the abstract already, because this is a subject of great current interest.
RESPONSE: We have done our best to describe the multi-pronged approach to transitioning off these drugs and now include information about tapering the dose or prolonging time between doses. Unfortunately, at this time, discontinuation is not a priority for clinicians and one of our goals in this paper is to bring attention to the need to consider the positives of doing so.
- COMMENT #2: l. 52 “the lack of long-term clinical trials” and “unforeseen long term side effects” are of course correct, but on the other hand the GLP-1 Ras have been on the market since 2005, and some of the very large studies (CVOTs and others) extend for up to 4-5 years, which are quite remarkable durations. The authors disregard the information that has accumulated during the most recent years.
RESPONSE: We have modified line 52 to include “In order to present these drugs with a balance of their pros and cons, the longer term studies showing cardiovascular benefit are also taken into account.” And added discussion and references to CVOTs in Section 4.3 GLP-1 Benefits to Organ Systems.
- COMMENT #3: 99: “(MASLD)…….is inflammation-driven” – not clear- it is driven by obesity, while MASH is inflammation. Fig.1 Is leptin “an inflammatory cytokine”?.
RESPONSE: We have corrected this with additional references (References 26 and 27). Leptin has multiple roles and is considered to be an inflammatory cytokine (La Cava A. Leptin in inflammation and autoimmunity. Cytokine. 2017;98:51-58. doi:10.1016/j.cyto.2016.10.011).
- COMMENT #4: Section 3.1 on Dietary approaches is superficial and contains nothing new
RESPONSE: Since diet is a critical means of adjusting weight and is clearly a part of any attempts to control weight with or without GLP-1 drugs, this section is key to our paper and we have added additional information.
- COMMENT #5: 163 The nice effects of Qsymia are not corrected for placebo
RESPONSE: According to Garvey et al: “At week 108, rates of weight loss were 1.8% for placebo, 9.3% for 7.5/46 mg, and 10.7% for 15/92 mg. Significantly more treated patients lost 5%, 10%, 15%, and 20% or more weight compared with those receiving placebo.”
- COMMENT #6: Section 3.3 is also very superficial.
RESPONSE: Surgical treatment of weight loss is the alternative to GLP-1 drugs that can meet or exceed weight loss achieved with the drugs and is therefore an important method to include. We would be remiss if we omitted it. We have added more detail in response to comments from another reviewer.
- COMMENT #7: 195 The effects on food-intake and weight loss are said to be self-evident? How is that? (But the decision to look at them for obesity therapy might seem self-evident) (although it was not, because of the side effects prevented people from trying higher doses and more powerful agonists had to be developed)
RESPONSE: We have removed “self-evident” and changed the phrasing to “While largely successful as an anti-diabetic drug therapy, the effects on both reducing food intake and promoting weight loss in persons with diabetes and animal models prompted further study as an anti-obesity medication” and added 2 new references (References 87 and 88).
- COMMENT #8: 222 “it is perhaps a common misconception that most GLP-1 secretion is gut derived.” If you are talking about circulating GLP-1 this is actually the case. The quantity of GLP1 produced in the periphery is also a lot larger. Actually this section need rewriting. The rise after bariatric surgery is certainly derived from the gut. It is important to note that no connection has been found between the peripheral GLP-1 system and the central (the ppg neurons of the brain stem).
RESPONSE: We have rewritten this section (lines 236-247) with additional references (References 110 and 11).
- COMMENT #9: 253: how did the “SELECT trial demonstrate cardiovascular benefits of semaglutide beyond the weight loss”?
RESPONSE: We have added this information (lines 274-289).
- COMMENT #10: l. 261. There was a three year extension of the SCALE 1 trial; also note that the SELECT trial showed a weight loss plateau of 4 years’ duration.
RESPONSE: We have added this information (lines 338-341).
- COMMENT #11: l. 272, there is also a three year’s extension of SURMOUNT 1 with tirzepatide.
RESPONSE: We have added this information and additional reference (Reference 137).
- COMMENT #12: 293: exenatide was the first GLP-1RA to be approved, but work on the GLP-1 agonist had begun long before that – the use of stabilized forms of GL-1 had begun already in 1998. It was the renal elimination that was the real hurdle (the iv. half-life of exenatide is only 30 min)
RESPONSE: We have corrected this (line 324) and added a reference from 1998 (Reference 143: Ahrén B. Glucagon-like peptide-1 (GLP-1): a gut hormone of potential interest in the treatment of diabetes. Bioessays. 1998, 20, 642-651.).
- COMMENT #13: 326. The effects of the GLP-1RAs on nausea and vomiting are not related to the effects on gastric emptying but are due to interactions with brain receptors. It is true that side effects can be elicited in many patients if you overdose, but more importantly they can be mitigated by slow up-titration.
RESPONSE: We have changed this (lines 364-366) and added the appropriate reference (Reference 157).
- COMMENT #14: 345. Most important regarding the risk of pancreatitis is to mention the CVOTs where the risk was minimal or absent, not because “they are recent” but because they are large and of long duration.
RESPONSE: We have included this information (lines 382-383) and reference (Reference 166: Cao, C.; Yang, S.; Zhou, Z. GLP-1 receptor agonists and pancreatic safety concerns in type 2 diabetic patients: data from cardiovascular outcome trials. Endocrine 2020, 68, 518–525).
- COMMENT #15: 355. Most important regarding the lean body mass loss is the ratio between fat mass and lean body mass. This ratio is generally maintained during GLP-1RA induced weight loss, indicating that there is no specific loss of lean body mass. Importantly, however, for some patient groups any loss can be a problem.
RESPONSE: There is still a lot of controversy regarding this ratio and we have added discussion to the manuscript (lines 401-403) with 2 new references (References 183,184).
- COMMENT #16: The mechanism behind the increase in heart rate has now been elicited (Cardiovasc Res. 2024 Oct 14;120(12):1427-1441).
RESPONSE: We thank the reviewer and have incorporated this information (line 413-414) and reference (Reference 185).
- COMMENT #17: Regarding the side effects of the GLP-1RAs the authors should describe the findings in the recent publication in Nature Med, https://doi.org/10.1038/s41591-024-03412-w
RESPONSE: We appreciate this guidance and have added this description (lines 417-422) and reference (Reference 190).
We thank the reviewer and believe that the manuscript is improved as a result of their input. We hope you will agree, and decide in favor of accepting our report at this time.

Reviewer 2 Report
Comments and Suggestions for Authors
The methodology of the study is not clear. Is it a systematic review? If yes the appropriate methodology should be applied. Perhaps it better fits to scoping review, but it also should be followed by an appropriate methodology.
Abstract
Please delete the statement concerning obesity. Start from GLP-1 revolutionized the management of obesity. Add information concerning weight maintenance.
Introduction
The introduction should be on obesity-related complications and obesity therapy, and not on epidemiology and history. Please remove most of the chapter 2. And make it much shorter. Figure 1 is useless.
Dietary approaches should include the therapy effectiveness and it should be combined with physical activity.
The literature has covered the topic of GLP -1 function. It should be restricted to a few sentences.
The paper should present a table with GLP-1a showing time of intervention and obtained effects.
The paper should present a table summarizing studies concerning weight regain after GLP-1a discontinuation.
The paper should discuss the necessity of long-term therapy beyond 2 years and potential strategies of the decreasing intensity of therapy that should be addressed in future studies.
Author Response
We thank the reviewer for thoroughly scrutinizing our manuscript. As requested, we have revised the manuscript and addressed the specific comments of the reviewer. The revised sections are delineated in red in a marked copy of the manuscript text.
Below, we provide a point-by-point response to the reviewer’s comments.
Reviewer # 3 Comments and Responses
- COMMENT #1: The methodology of the study is not clear. Is it a systematic review? If yes the appropriate methodology should be applied. Perhaps it better fits to scoping review, but it also should be followed by an appropriate methodology.
RESPONSE: This is not a systematic review. We looked at many other reviews of this type in Biomolecules and none included a methodology.
- COMMENT #2: Abstract Please delete the statement concerning obesity. Start from GLP-1 revolutionized the management of obesity. Add information concerning weight maintenance.
RESPONSE: We have deleted the statement as requested.
- COMMENT #3: Section Introduction: The introduction should be on obesity-related complications and obesity therapy, and not on epidemiology and history. Please remove most of the chapter 2. And make it much shorter. Figure 1 is useless.
RESPONSE: We believe that this introduction is appropriate and not excessively long. We prefer to retain the figure.
- COMMENT #4: Dietary approaches should include the therapy effectiveness and it should be combined with physical activity.
RESPONSE: We have added the requested information (lines 134-138) and references 45 and 54.
- COMMENT #5: The literature has covered the topic of GLP-1 function. It should be restricted to a few sentences.
RESPONSE: We intend for this review to serve as a resource for persons from multiple disciplines and believe that it is important to provide context within the paper itself for those who are not experts in the field.
- COMMENT #6: The paper should present a table with GLP-1a showing time of intervention and obtained effects.
RESPONSE: We have added this table (Table 2).
- COMMENT #7: The paper should present a table summarizing studies concerning weight regain after GLP-1a discontinuation.
RESPONSE: We have added this table (Table 3).
- COMMENT #8: The paper should discuss the necessity of long-term therapy beyond 2 years and potential strategies of the decreasing intensity of therapy that should be addressed in future studies.
RESPONSE: We now include these elements in the discussion under “Present and Future of Weight Loss.”
We thank the reviewer and believe that the manuscript is improved as a result of their input. We hope you will agree, and decide in favor of accepting our report at this time.

Round 2
Reviewer 1 Report
Comments and Suggestions for Authors
No further comments
Author Response
We thank the reviewer for thoroughly scrutinizing our manuscript.
Below, we provide a point-by-point response to the reviewer’s comments.
Reviewer # 1 Comments and Responses
- COMMENT: No further comments
RESPONSE: We appreciate your time and effort in reviewing our manuscript.
We thank the reviewer once again and believe that the manuscript is improved as a result of their input. We hope you will agree, and decide in favor of accepting our report at this time.

Reviewer 2 Report
Comments and Suggestions for Authors
The paper was improved. However, the methodology aspect is still not clear.
In my opinion, it should be done in accordance with the Scoping—PRISMA statement. The registration of the protocol is not the most important aspect.
If this aspect is solved, the paper is ready for publication.
Author Response
We thank the reviewer for thoroughly scrutinizing our manuscript. As requested, we have revised the manuscript and addressed the specific comments of the reviewer. The revised sections are delineated in red in a marked copy of the manuscript text.
Below, we provide a point-by-point response to the reviewer’s comments.
Reviewer # 3 Comments and Responses
- COMMENT: The paper was improved. However, the methodology aspect is still not clear. In my opinion, it should be done in accordance with the Scoping—PRISMA statement. The registration of the protocol is not the most important aspect. If this aspect is solved, the paper is ready for publication.
RESPONSE: This is not a scoping review; it is a narrative review. We have now added a methodology section (Section 2) describing the process through which we prepared the manuscript.
We thank the reviewer and believe that the manuscript is improved as a result of their input. We hope you will agree and decide in favor of accepting our report at this time.
